# Comparative Analysis of Biological Signatures between Freshly Preserved and Cryo-Preserved Bone Marrow Mesenchymal Stem Cells

**DOI:** 10.3390/cells12192355

**Published:** 2023-09-26

**Authors:** Taesic Lee, Sangwon Hwang, Dongmin Seo, Sungyoon Cho, Sunja Yang, Hyunsoo Kim, Jangyoung Kim, Young Uh

**Affiliations:** 1Division of Data Mining and Computational Biology, Regenerative Medicine Research Center, Wonju Severance Christian Hospital, Wonju 26426, Republic of Korea; ddasic123@yonsei.ac.kr; 2Department of Family Medicine, Yonsei University Wonju College of Medicine, Wonju 26426, Republic of Korea; 3Department of Precision Medicine, Yonsei University Wonju College of Medicine, Wonju 26426, Republic of Korea; arsenal@yonsei.ac.kr; 4Department of Medical Information, Yonsei University Wonju College of Medicine, Wonju 26426, Republic of Korea; dmseo@yonsei.ac.kr; 5Pharmicell Co., Ltd., Seongnam 13229, Republic of Korea; sungyoon9030@pharmicell.com (S.C.); sun-ja@pharmicell.com (S.Y.); khsmd@pharmicell.com (H.K.); 6Department of Internal Medicine, Yonsei University Wonju College of Medicine, Wonju 26426, Republic of Korea; kimjy@yonsei.ac.kr; 7Department of Laboratory Medicine, Yonsei University Wonju College of Medicine, Wonju 26426, Republic of Korea

**Keywords:** mesenchymal stem cell, cryo-preserved, freshly preserved, bone marrow, biological signature, paracrine molecules

## Abstract

Mesenchymal stem cells (MSCs) can differentiate into multiple connective tissue lineages, including osteoblasts, chondrocytes, and adipocytes. MSCs secrete paracrine molecules that are associated with immunomodulation, anti-fibrotic effects, and angiogenesis. Due to their orchestrative potential, MSCs have been therapeutically applied for several diseases. An important aspect of this process is the delivery of high-quality MSCs to patients at the right time, and cryo-biology and cryo-preservation facilitate the advancement of the logistics thereof. This study aimed to compare the biological signatures between freshly preserved and cryo-preserved MSCs by using big data sourced from the Pharmicell database. From 2011 to 2022, data on approximately 2300 stem cell manufacturing cases were collected. The dataset included approximately 60 variables, including viability, population doubling time (PDT), immunophenotype, and soluble paracrine molecules. In the dataset, 671 cases with no missing data were able to receive approval from an Institutional Review Board and were analyzed. Among the 60 features included in the final dataset, 20 were selected by experts and abstracted into two features by using a principal component analysis. Circular clustering did not introduce any differences between the two MSC preservation methods. This pattern was also observed when using viability, cluster of differentiation (CD) markers, and paracrine molecular indices as inputs for unsupervised analysis. The individual average PDT and cell viability at most passages did not differ according to the preservation method. Most immunophenotypes (except for the CD14 marker) and paracrine molecules did not exhibit different mean levels or concentrations between the frozen and unfrozen MSC groups. Collectively, the biochemical signatures of the cryo-preserved and unfrozen bone marrow MSCs were comparable.

## 1. Introduction

The concept of stem cells originated approximately 150 years ago as a theoretical postulate describing the self-renewal ability of certain tissues in an organism throughout its lifetime [1]. Tavassoli and Crosby [2] demonstrated that bone marrow (BM) transplantation into heterotopic anatomical regions resulted in de novo ectopic bone and BM generation. Friedenstein et al. [3] reported that a minor subpopulation of BM cells had a fibroblast-like appearance and that adherence to the vessel was associated with osteogenic potential; therefore, Friedenstein is generally credited for the discovery of osteogenic or BM stromal stem cells [4], which are officially termed mesenchymal stem cells (MSCs) [5]. MSCs have the capacity for multilineage differentiation [6] and mainly undergo trilineage differentiation into multiple connective tissue lineages, such as osteoblasts, chondrocytes, and adipocytes [7]. MSCs can be obtained from various sources, including BM, adipose tissue, dental pulp, umbilical cord blood, amniotic fluid, placenta, peripheral blood, synovium, endometrium, skin, muscle, pancreas, and liver [8,9]. Because it involves less invasive separation procedures than those for the extraction of stem cells from organs, such as the pancreas and liver, along with the substantial accumulation of its proven therapeutic effects, BM tissue has become the standard source of MSCs worldwide [8,9].

Accordingly, MSCs can theoretically be used in regenerative medicine (bone, cartilage, tendon, skeletal muscle, myocardium, and neural tissue regeneration) and immunomodulatory (acute graft-versus-host disease and autoimmunity) and anti-cancer cell therapies [5,10,11]. Specifically, BM-MSCs have been therapeutically applied to a variety of clinical conditions, such as liver disease [11] and cardiovascular disease [12], due to their immunomodulatory, anti-apoptotic, and angiogenic properties [13]. In Korea, Pharmicell (Pharmicell Co., Ltd., Seoul, Republic of Korea) provided the first globally commercialized stem cell therapy (Hearticellgram-AMI) approved by a government regulatory institute (Korea Food and Drug Administration) [14]. The future cell bank (FCB) Pharmicell has partnered with several medical centers in Korea to validate the therapeutic effects of BM-MSC manufactured in its Good Manufacturing Practice (GMP) facility on a wide range of diseases, such as stroke, spinal cord injury, liver cirrhosis, kidney disease, erectile dysfunction, and critical limb ischemia. By using Pharmicell and GMP as keywords, numerous published studies, such as clinical trials, in vitro experiments, and in vivo experiments, were collected. Among them, 12 clinical studies, including 5 randomized clinical trials (RCTs), 4 open-label trials, and 3 case studies, were reviewed. A list of these studies is presented in Table 1.

Crucial aspects of stem cell therapy are the provision of consistently good-quality and reliable MSCs and the transport of cell materials to patients in a timely manner. Cryo-biology facilitates successful cell therapy by enabling the long-term storage of consistently efficient stem cells and their origin tissues. However, cryo-preservation has several potential limitations, such as the functional alteration of stem cells and decreased therapeutic potential [27,28]. On the contrary, several studies suggested that the functionality of cryo-preserved MSCs was comparable to that of freshly preserved MSCs [29]. Of the 12 studies reviewed in Table 1, 2 studies that successfully validated the anti-fibrotic effect of BM-MSCs on alcoholic liver cirrhosis used both fresh and cryo-preserved stem cells and did not report any differential clinical results related to the preservation method. The FCB Pharmicell has performed numerous stem cell therapies for acute or chronic diseases, producing big data that are helpful for obtaining generalized findings. All eligible data were collected and preprocessed into structured and normalized datasets. Then, this study aimed to compare the biological signatures between freshly preserved and cryo-preserved MSCs based on the representative stem cells’ viability, proliferative duration, immunophenotypes, and soluble paracrine molecules. We conducted a database- and statistics-based analysis of commercially generated MSC products to evaluate the differential biochemical signatures of BM-MSCs according to the two preservation methods (Figure 1).

## 2. Materials and Methods

### 2.1. Data Preparation

Data on 2300 stem cell therapies from 2011 to 2022 were collected from the Pharmicell database. The dataset was established to include approximately 60 features, such as hospitals where BM was aspirated and stem cell therapy was performed, manufacturing time, patient demographic data, population doubling time (PDT), viability, cluster of differentiation (CD) markers, and cellular biochemistry indices. All participants were de-identified; therefore, this study excluded identifiable information of the enrolled participants. Moreover, this dataset did not include medical information such as past medical history of chronic diseases and the reason for receiving stem cell therapy.

The following inclusion criteria were used: (1) eligibility to proceed with the approval of an Institutional Review Board; (2) completeness of data. First, the samples obtained from Dr. Kim’s Stem Cell Clinic and Wonju Severance Christian Hospital were eligible for inclusion. Then, cases with missing data on the variables that were compiled were eliminated, yielding 671 cases for the final analysis. All BM-MSCs analyzed in the current study were derived exclusively from autologous cell therapy cases.

This study was approved by the Public Institutional Review Board Designated by Ministry of Health and Welfare of South Korea (IRB number: P01-202307-01-016). This study was performed in accordance with the principles of the Declaration of Helsinki.

### 2.2. Preparation and Preservation of BM-MSCs

The procedures performed to generate clinical-grade BM-MSCs were as follows: BM aspiration, isolation of mononuclear cells (MNCs), cell culture of MSCs, and evaluation of the quality of BM-MSCs for clinical usage (Figure 1A). MNCs isolated through the centrifugation of BM aspirates were categorized into two groups: freshly preserved and cryo-preserved MNCs (Figure 1B). The cryo-preserved MNCs were dispensed in 1 mL storage vials, placed in a cryovial container, and stored in a controlled-rate freezer for approximately 4 h. Subsequently, the initially frozen MNCs were transferred to a vial storage tank filled with liquid nitrogen until stem cell therapy (Figure 1B). The cryo-preserved MNCs were thawed in a thermal water bath at 37 °C with gentle and constant shaking (Figure 1B and Appendix A). Post-thawing, MNCs were washed and plated for cell culture (Figure 1 and Appendix A). In the freshly preserved group, MNCs were directly processed into culture systems.

Appropriate quantities of MNCs were inoculated into a T75 flask containing low-glucose Dulbecco Modified Eagle Medium (DMEM) supplemented with 10% FBS and cultured in an incubator at 37 °C in a 5% CO_2_ atmosphere. After 5–7 days of cultivation, only the BM-MSCs attached to the flask wall were separated and inoculated into a new flask, establishing passage 0 (P0). Depending on the saturation of stem cells (70–80% confluence), MSCs were serially subcultured every 3–5 days, leading up to P4 or P5. After the aforementioned process, MSCs were harvested for the final pharmaceutical production.

BM-MSCs’ quality was evaluated based on the following criteria: an MSC viability of greater than 70%; absence of bacteria, fungi, viruses, and *Mycoplasma*; more than 90% of cells expressing CD73 and CD105; less than 3% of cells expressing CD14, CD34, and CD45 [23]. The manufacturing practices have been described in detail in previous studies [20,21,23].

### 2.3. Biological Signatures

A trypan blue exclusion assay was performed to assess the viability of BM-MSCs, and the cells were counted by using a hemocytometer under a light microscope. The cell counting with a hemocytometer was conducted three times, and the average was determined as the final MSC count. The viability of the MSCs was evaluated at least six times from passages 0 to 4 (P0–4) and the final product. Some cases included information on P5; however, this study only analyzed data obtained from P0 to P4 and the final product.

The PDT of the BM-MSCs was evaluated by using the Patterson formula. In detail, the PDT was calculated by using the following equation:$$\mathrm{PDT}=T\times \ln2/\ln\frac{N\left(t\right)}{N_{0}}$$
where *T* is the incubation time, *N*(*t*) is the number of cells at *t* hours of culture, and *N*_0_ is the number of cells seeded at the initial time point. The PDT was evaluated at every passage of the cell culture.

Immunophenotypic analysis was conducted by staining cells with five antibodies conjugated to fluorescein isothiocyanate (FITC) or phycoerythrin (PE): anti-CD14-FITC, anti-CD34-FITC, anti-CD45- FITC, anti-CD73-PE, and anti-CD105-PE (BD Biosciences, San Jose, CA, USA). Flow cytometry (Navios, Beckman Coulter, Brea, CA, USA) was used to evaluate the cell fluorescence intensity.

MSCs exert therapeutic effects by secreting anti-apoptotic (vascular endothelial growth factor (VEGF), hepatocyte growth factor (HGF), and insulin-like growth factor 1), immunomodulatory (prostaglandin E2, transforming growth factor beta, and HGF), and angiogenic (VEGF, monocyte chemoattractant protein 1 (MCP1), and interleukin 6 (IL6)) molecules [13]. Kwon et al. [30] demonstrated that VEGF, MCP1, and IL6 are the crucial factors responsible for MSC-related angiogenesis. Based on cumulative evidence [13,30], the combination of VEGF, MCP1, and IL6 was determined as an index of therapeutic angiogenesis. VEGF, MCP1, and IL6 concentrations were measured by using enzyme-linked immunosorbent assay kits according to the manufacturer’s protocol (R&D Systems, Minneapolis, MN, USA). A subset of MSCs were collected during the immediate pre-production phase of final pharmaceutical production. MSCs were then cultured for 72 h in media under the same conditions as those of the main culture without any additional stimulation or treatment. Finally, ELISA-based cytokine levels were measured in culture supernatants.

### 2.4. Statistics

Continuous variables, such as age, cell counts, PDTs, CD markers, and chemokines, were analyzed by using Student’s *t*-test to identify the differential signatures between the two preservation methods.

Principle component analysis (PCA) was used to reduce the dimensionality of the biochemical features of BM-MSCs. Univariate and multivariate linear regressions were used to examine the differential biochemical signatures between the freshly preserved and cryo-preserved groups. All statistical analyses were conducted with the R language (version 4.0.1, R Foundation for Statistical Computing, Vienna, Austria).

## 3. Results

### General Characteristics of BM-MSCs

Of the 2290 cases commissioned from over 100 hospitals, only 715 from 2 hospitals (Kim’s Clinic and WSCH) were able to go through the IRB approval process (Figure 2). Among the 715 MSC cases, the general characteristics of the 671 MSCs with complete biological data stratified according to the preservation method are presented in Figure 2 and Appendix A. The donor age in the cryo-preserved MSC group was lower than that in the freshly preserved MSC group. As the first step of MSC production, 6–60.5 mL of BM was aspirated from the posterior superior iliac crest of patients scheduled for stem cell therapy under local anesthesia (Appendix A). The distribution of aspirated BM was similar to the log-normal pattern (Appendix A). The amount of aspirated BM in the cryogenically preserved MSC group was less than that in the freshly preserved group (Figure 2). The average duration between manufacturing and the completion of BM-MSC processing and stem cell therapy was 28.2 days (Appendix A) and did not differ according to the BM-MSC preservation method (Figure 2). The numbers of seeded MNCs and MSCs at P0, P1, and P4 were similar between the freshly and cryo-preserved MSC groups (Appendix A).

A multidisciplinary team that included experts in GMP-facility-based MSC production and a data scientist who preprocessed the dataset that was extracted from the company database selected approximately 20 variables (exactly 21 features) as representative markers characterizing stem cells, such as donor age, PDT, viability of MSC, CD markers, VEGF, MCP1, and IL6. The data experts selected features with the top 80, 70, 60, and 50 percent for the highest standard deviations across the stem cell cases. Four combinatory lists of abstract variables summarized from the four lists (i.e., top 80, 70, 60, and 50 features) of biochemical indices through PCA showed a circle-shaped clustering, indicating a low probability of the presence of subgroups with differential characteristics (Appendix A). These patterns were also shown in the embedding features (PCs) obtained from all variables (n = 21) representing BM-MSC status (Figure 3). Moreover, the task of dimensionality reduction did not distinguish between the cryo- and freshly preserved MSC groups (Figure 3). When reducing the six dimensions of viability (viabilities in P0 to P4 and the final product’s viability) into two by using PCA, a triangle-shaped cluster was established, and this did not include any differential groups according to the preservation method. When PCA was applied to CD markers and therapeutic-effect-related indices (VEGF, MCP1, and IL6), triangle-shaped clusters were constructed (referred to as “CD markers” and “VEGF, MCP1, IL6” in Figure 3) and did not show classified patterns between frozen and unfrozen MSCs.

The mean PDT values for P0, P1, P2, P3, and P4 were 61.3, 61.8, 66.1, 68.6, and 93.4 h, respectively. The PDT at each passage (P0–P4) did not differ between the freshly and cryo-preserved MSC groups (Figure 4). The average donor age was significantly lower in the cryo-preserved MSC group than in the unfrozen MSC group (Figure 2 and Appendix A). Therefore, multivariate analyses adjusted for age were conducted. The results showed no significant differences in PDT in every passage between the freshly and cryo-preserved MSC groups (Figure 4).

The minimum–maximum levels of viability were 45.9–100%, 82.5–100%, 90.1–100%, 84.8–100%, and 72.8–100% in P0, P1, P2, P3, and P4, respectively. The viability at P1 did not differ between the freshly and cryo-preserved MSC groups (Figure 5). After adjusting for age, the viability at P1 was not significantly different between the freshly and cryo-preserved MSC groups (Figure 5). At every passage (P0–P4), the two MSC preservation groups exhibited high viability without significant differences in cell viability in the univariate and multivariate regression analyses (Figure 5), strongly supporting the finding in Figure 3 that the combinatory signatures of viability did not differ between the preservation methods.

The CD105 and CD73 markers of MSCs exhibited high ratios without differences between the freshly and cryo-preserved groups (Figure 6). Insignificant differences in CD105 and CD73 levels remained after adjusting for donor age. For CD34, CD45, and CD14, all MSCs showed extremely low values (less than 3 percent). The difference in CD34 and CD45 expression in BM-MSCs between the fresh- and cryo-preserved groups was insignificant in both the univariate and multivariate analyses. In addition, the average CD14 expression of MSCs in the cryo-preserved group was higher than that in the freshly preserved group, and the difference was significant in both the univariate and multivariate linear regressions (Figure 6).

Kim et al. [31] reported that the concentration of VEGF in transfected human MSCs ranged from 1000 to 2500 pg/mL; our study reported a similar distribution of VEGF concentration. In our study, the levels of MCP1 (21.6–119.7 pg/mL) secreted by umbilical cord blood MSCs were lower than those secreted by BM-MSCs [32]. Choi et al. [33] reported that the concentrations of IL6 in BM-MSCs at passages 3, 5, 7, and 9 were approximately 220, 190, 140, and 120 pg/mL, respectively, which were lower than those secreted by BM-MSCs (1888 pg/mL in unfrozen and 1810 mg/mL in frozen MSCs). No significant differences were observed in the concentrations of VEGF, MCP1, and IL6 between the freshly and cryo-preserved groups before and after adjusting for age (Figure 7).

Finally, a network analysis was conducted to describe the 21 features employed for characterizing BM-MSC cases in the present study (Figure 8). Correlation matrices were constructed for all freshly preserved and cryopreserved MSCs (Figure 8A, Figure 8B and Figure 8C, respectively) by calculating Pearson’s correlation coefficient (PCC) for all possible pairs of variables. Subsequently, three networks were constructed based on the interactions between two features that showed a PCC of 0.2 or higher. There were positive correlations among VEGF, MCP1, and IL6, and these were consistently observed across all three networks (Figure 8). Dependency between the PDTs of P3 and P4 was observed in cryopreserved BM-MSCs, but not in other networks. Most of the interaction patterns were similar among the three preservation-specific BM-MSC networks.

## 4. Discussion

This study analyzed the differential biochemical signatures of BM-MSCs according to preservation methods—namely, the fresh preservation and cryo-preservation methods—and found that BM-MSCs did not exhibit variations in characteristics between the freshly and cryo-preserved groups. Based on the dimensionality reduction method (i.e., PCA), a circular clustering shape was obtained, indicating that BM-MSCs were highly homogenous and could not be classified between the two types of preservation methods. The highly homogenous characteristics of MSCs observed in our study could be a cornerstone of the transition from autologous to allogeneic stem cell therapy.

The PDTs for most passages, except for P4, did not differ between the freshly and cryo-preserved BM-MSC groups. Oja et al. [35] conducted a comparative analysis among unfrozen MSCs, interim frozen MSCs at P0, and interim frozen MSCs at P1, and they found no differences in PDTs. In addition, the repeated cryo-preservation of MSCs can significantly increase PDTs [36,37,38]. In P4, the cryo-preserved stem cells required a shorter duration to double their number, indicating that the cryo-effect completely disappeared in the late phase of passage.

The viability of BM-MSCs, which is an important attribute of cell functionality, did not exhibit significant differences between the unfrozen and cryo-thawed MSC groups at any passage. Some studies have suggested further elucidation of the effect of cryo-preservation because they found that the freeze–thaw process could affect MSC function [39,40]. In addition, Ginis et al. [29] observed that MSCs that underwent hypothermic storage for four days did not show changes in viability, proliferation, or differential potential. A systematic review reported that the post-thaw viability of MSCs varied from 50 to 100% after reviewing 37 relevant studies; of 26 human MSC studies, 16 did not report any changes in post-thaw viability, thus supporting our findings [41].

A comparative analysis of immunophenotyping did not show significant differences between the two preservation methods (Figure 3 and Figure 6). However, cryo-preserved BM-MSCs showed higher CD14 levels than freshly preserved stem cells did. CD14, which is mainly expressed by neutrophils, macrophages, and dendritic cells, is involved in the activation of lipopolysaccharides and the Toll-like receptor 4 signaling pathway [42], and its deficiency is considered to be a minimal criterion for defining multipotent MSC identity [43]. A recent study reported that a small fraction of MSCs (<5%) expressed CD14 [44], and the criterion of clinical usage of BM-MSCs is a CD14 expression of less than 3% [21,23]. Although frozen–thawed MSCs exhibited higher CD14 levels than those of fresh cells, their level of CD14 expression satisfied the criteria for clinical usage.

VEGF, MCP1, and IL6, the therapeutic indices of MSCs in the cryo-preserved group, were well preserved compared to those in the freshly preserved group. MSCs exhibit therapeutic effects, such as immunomodulation, anti-apoptosis, angiogenesis, anti-fibrosis, and chemo-attraction, by secreting paracrine factors (referred to as the secretome) [13,45]. Kim et al. [46] identified a combination of paracrine factors, including angiogenin, IL8, MCP1, and VEGF, as therapeutic-effect-related markers. Kwon et al. [30] reported a subset of paracrine factors, such as VEGF, MCP1, and IL6, as stem-cell-therapy-related markers. MSCs encompass a novel secretome network that cannot be substituted by single chemokine agents [45]; therefore, co-expression network analysis or deep learning is required to discover MSC-prognosis-related biomarkers [47,48].

Traditional metrics used to define MSC identity include plastic adherence, fibroblast-like morphology, cell surface phenotyping, and tri-lineage differentiation [43]. Moreover, an attempt to evaluate the predictive therapeutic effect of MSCs was made based on molecules secreted by stem cells, such as VEGF, MCP1, and IL6 [30]. In our study, although most stem cells were homogeneous when evaluated by using the aforementioned indices, a non-negligible number of MSCs existed topologically on the periphery of the cluster (Figure 3). To establish the intrinsic properties of MSCs, the deep phenotyping of MSCs was performed [49]. Several studies have attempted to uncover MSC-related molecular signatures by using transcriptomic analysis [50,51]. Gupta et al. [52] recently analyzed the mRNAs of all genes isolated from BM-MSCs and identified several therapy-response-related genes. Moreover, we previously established an analytical platform for transcriptome data based on a co-expression network [47], which can be used in future studies to determine molecular signatures related to the identity and therapeutic effects of MSCs.

Many researchers are concerned about cryopreservation potentially inducing detrimental changes in stem cells [39,53,54]. Artebi et al. [53] suggested that factors such as the cytotoxicity caused by cryoprotectants (DMSO), the freezing duration, and the freezing method have an adverse impact on MSCs. Furthermore, they reported transcriptomic alterations in MSCs subjected to freezing and thawing processes [53]. Other studies have shown that cryopreservation affects protein translation in MSCs [54]. The aforementioned studies, which highlight the adverse impacts of the freezing process on stem cells, commonly show changes in cellular-culture-related indices, such as viability and PDT, during subsequent passages [39,53,54]. However, in our study, no significant differences were observed between frozen and fresh cells in terms of PDT, viability (except for P1), or other cellular-cultivation-related indices during most passages in the culturing process. Cryopreservation is an essential method for maintaining MSC properties and preventing cellular senescence. Several studies have reported that the cultivation-related characteristics of MSCs can be preserved even with long-term cryopreservation for up to 6 months [4] and that various properties, including morphology, telomerase activity, karyotype profile, expression of CD markers, proliferation rate, and differentiation potential, are maintained even when stored for 10 to 12 months in a nitrogen tank [5]. The current study did not include an analysis of the storage duration in nitrogen tanks. However, when comprehensively analyzing the global signatures of BM-MSCs by using dimensionality reduction techniques, a homogeneous pattern where MSCs converged in a single cluster occurred, and no discernible differences were observed between cryo- and freshly preserved MSCs. This indicates that cells possess homogenous characteristics independent of the preservation method and duration of LN2 storage. The current study did not include an analysis of the storage duration in nitrogen tanks. To calculate the duration of storage in an LN2 tank, it is necessary to verify the date of BM extraction and of obtaining passage 0. However, this information includes details that allow the tracing of the origin of MSCs back to the specific individuals from whom they were collected. Due to privacy issues and the approval of the ethics committee, we could not extract the aforementioned information from the company database. Furthermore, when comprehensively analyzing the global signatures of BM-MSCs by using dimensionality reduction techniques, a homogeneous pattern where MSCs converged in a single cluster occurred, and no discernible differences were observed between cryo- and freshly preserved MSCs. This indicates that cells possess homogenous characteristics independent of the preservation method and duration of LN2 storage.

From P1 to P4, an increase in PDT was observed, which is a phenomenon of cellular senescence, a common occurrence in cell culture that is also referred to as in vitro aging [55]. A recent study attempted to use alternative materials, such as αMEM (minimum essential medium alpha), instead of DMEM to attenuate in vitro aging [55]. Our research is a database- and statistics-based analysis of commercially generated MSC products using an established and standardized platform (Figure 1). Some MSCs exhibit initial signs of in vitro aging; however, no additional interventions have been implemented to alleviate cellular senescence. Instead, by harvesting MSCs at an early stage—specifically, at passages 4–5—with an adequate injection volume, the final product was generated before the onset of or during the early stages of in vitro aging.

The global characterization of cytokines is challenging in clinical settings when evaluating the immunomodulatory properties of BM-MSCs. Liu et al. [56] measured the concentrations of IL4, IL6, IL10, IL13, TNFα, and VEGF to evaluate MSCs’ immunomodulatory potential. To perform the ELISA-based profiling of cytokines, additional laboratory studies, such as further cell culturing and the induction of antigen–antibody reactions, are required [57]. Evaluating the differentiation capability of MSCs requires approximately 1 month of additional experiments and cell staining [57]. Therefore, to assess the differentiation potential of MSCs, transcriptomic profiling has been implemented for a genome-wide range [58], as well as for targeted genes, such as *Col I*, *RUNX2*, *ALP*, *LPL*, and *PPARγ* [55]. The current study has a retrospective design and involves database-based research analyzing accumulated data by using machine learning and statistical methods, rather than experiment-based functional studies. Initiated by the present study, a future experimental study is planned to be conducted, and our team is in the process of establishing an analysis platform for biomarker discovery, which will include differential and co-differential patterns [47,59]. Future research utilizing big data, deep phenotypes (e.g., experimental indices and genetic/transcriptomic profiles), and the aforementioned analytical platforms is required to identify in-depth and generalized MSC-related biosignatures.

## 5. Conclusions

We implemented realistic big data from a stem cell therapy company and concluded that the biosignatures of cryo-preserved BM-MSCs were highly conserved with respect to those of unfrozen BM-MSCs.

## Figures and Tables

**Figure 1 cells-12-02355-f001:**
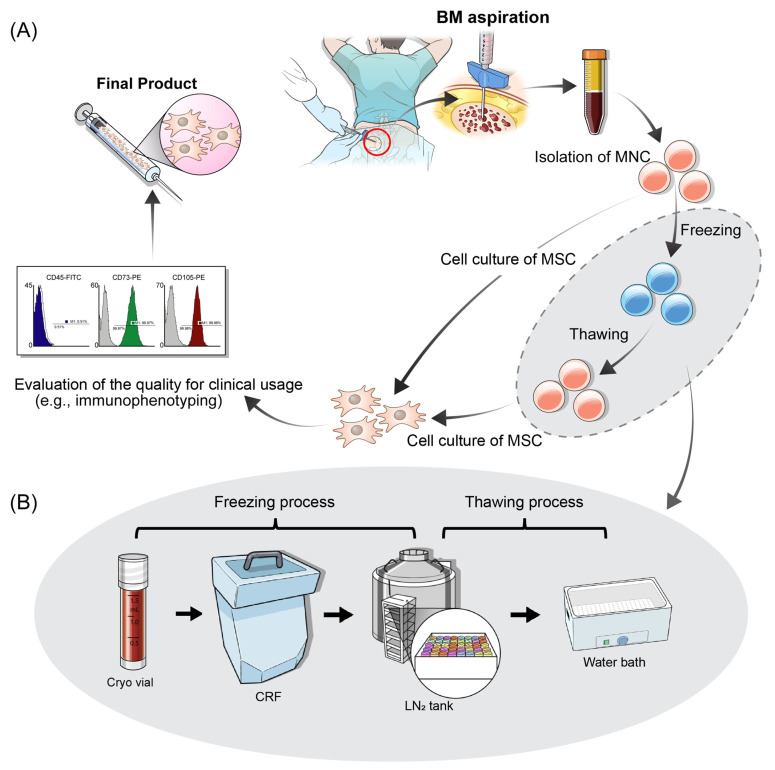
Processing of the production of BM-MSCs. (**A**) Four representative steps are required to obtain the final product of BM-MSCs, including the aspiration of BM, isolation of MNCs, culturing of MSCs, and evaluation of the MSCs’ quality. (**B**) A modest fraction of MNCs are arranged as the cryo-preserved group and subsequently undergo freezing and thawing. The freezing process is performed in two steps and requires the use of a CRF (for about 4 h) and nitrogen tank (semi-permanent). Thawing is manually conducted by using a water bath. This figure illustrates a schematic plot, and an actual photograph is presented in Appendix A. Abbreviations: BM-MSC; bone-marrow-derived mesenchymal stem cell; MNC, mononuclear cell; CRF, controlled-rate freezer; LN_2_, liquid nitrogen.

**Figure 2 cells-12-02355-f002:**
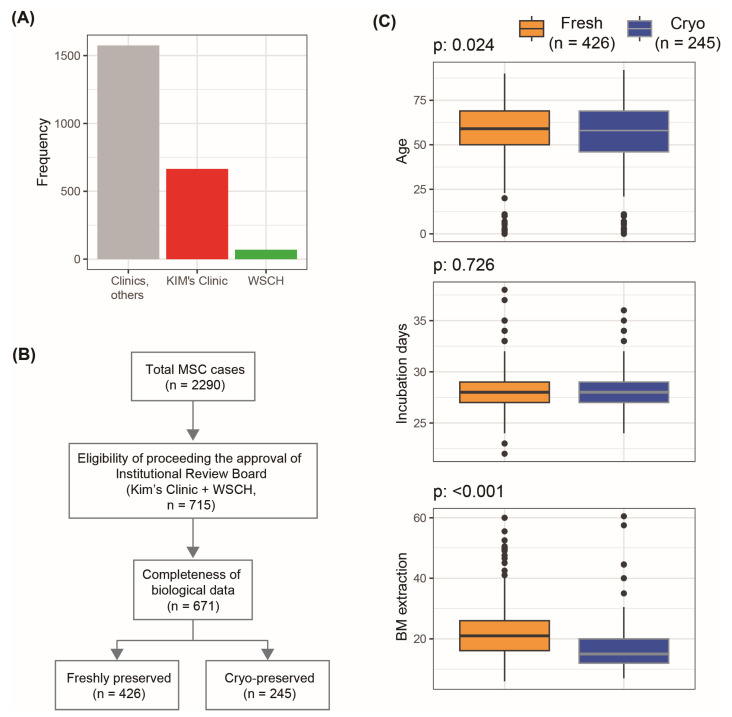
The process of determining the number of analyzable BM-MSC cases and their general characteristics according to the preservation method. (**A**) Stem cell therapies were conducted at various hospitals, among which autologous BM-MSC cases from two hospitals (Kim’s Clinic and WSCH) could undergo the IRB approval process. (**B**) Two steps of MSC case selection were conducted to provide all of the results in the overall analysis platform of the current study. (**C**) The differential patterns of variables of stem cells according to the preservation methods were evaluated based on Student’s *t*-test. Abbreviations: BM, bone marrow; Cryo: cryopreservation; Fresh: freshly preserved MSCs; MSC, mesenchymal stem cell; WSCH, Wonju Severance Christian Hospital.

**Figure 3 cells-12-02355-f003:**
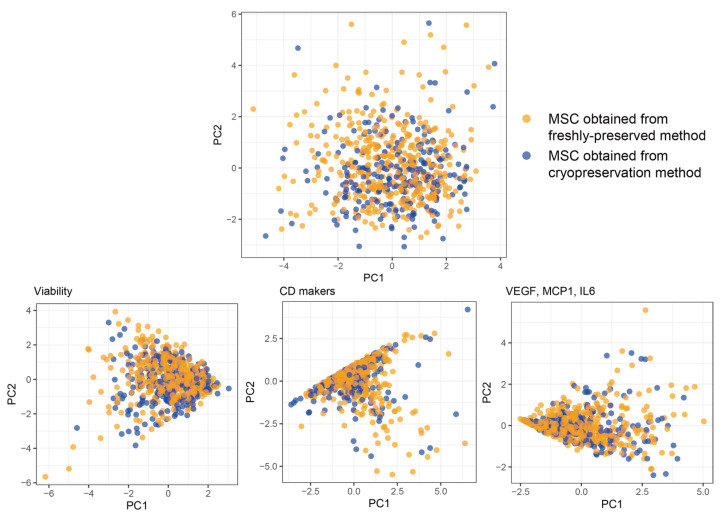
Comparative analysis of the biological signatures between the freshly and cryo-preserved MSC groups. The viability includes values measured at P0–4 and at the final product time. The CD markers included CD14, CD34, CD45, CD73, and CD105 and were measured by using flow cytometry. The lower-right plot shows the distributions of PC1 and PC2 abstracted from three molecules, including VEGF, MCP1, and IL6. The plot labeled “overall” indicates the distribution of dimensionality-reduced features, including viability, CD markers, paracrine molecules (VEGF, MCP1, and IL6), BM amount, PDT, and donor’s age, assessed by using PCA. Abbreviations: PC, principal component; VEGF, vascular endothelial growth factor; MCP1, monocyte chemoattractant protein 1; IL6, interleukin 6.

**Figure 4 cells-12-02355-f004:**
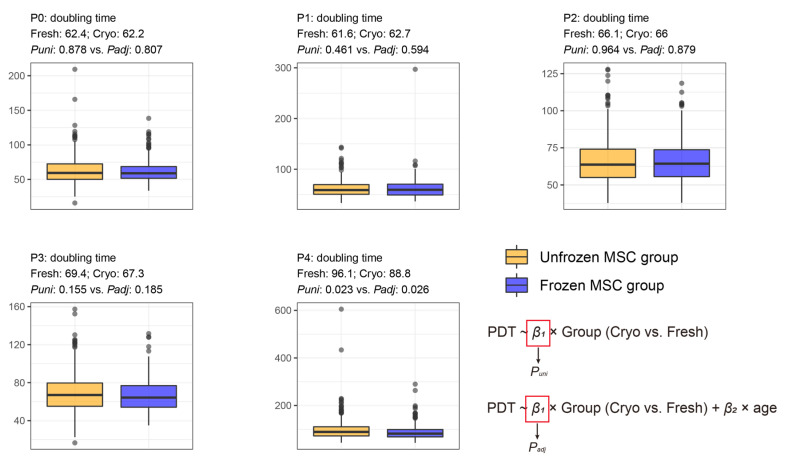
Comparative analysis of the PDT at every cell culture passage between freshly preserved and cryo-preserved MSC groups. “Fresh: *m* and Cryo: *n*” indicates *m* and *n* of the average feature values in the freshly preserved and cryo-preserved MSC groups, respectively. *P_uni_* and *P_adj_* indicate *p*-values that were evaluated by using univariate and multivariate (adjusting age) linear regression analyses, respectively. The median and inter-quartile range are described in Appendix A.

**Figure 5 cells-12-02355-f005:**
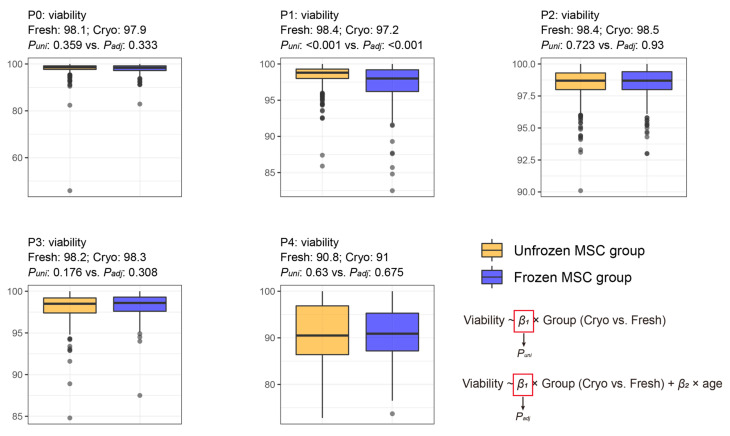
Comparative analysis of the viability at every cell culture passage between the freshly preserved (state 0) and cryo-preserved (state 1) MSC groups. “Fresh: *m* and Cryo: *n*” indicates *m* and *n* of the average feature values in the freshly preserved and cryo-preserved MSC groups, respectively. *P_uni_* and *P_adj_* indicate *p*-values that were measured by using univariate and multivariate (adjusting for age) linear regression analyses. The median and interquartile range are described in Appendix A.

**Figure 6 cells-12-02355-f006:**
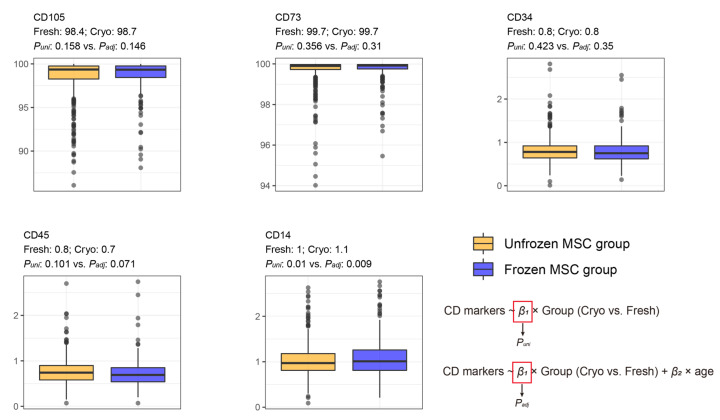
Comparative analysis of CD markers between freshly and cryo-preserved MSC groups. “Fresh: *m* and Cryo: *n*” indicates *m* and *n* of the average feature values in the freshly preserved and cryo-preserved MSC groups, respectively. *P_uni_* and *P_adj_* indicate *p*-values that were evaluated by using univariate and multivariate (adjusting age) linear regression, respectively. Detailed values for the median and interquartile range are described in Appendix A.

**Figure 7 cells-12-02355-f007:**
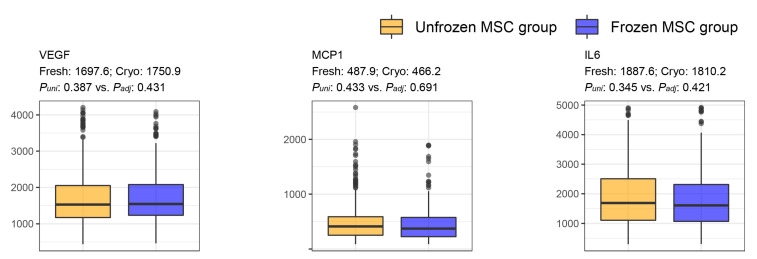
Comparative analysis of VEGF, MCP1, and IL6 between the freshly and cryo-preserved MSC groups. “Fresh: *m* and Cryo: *n*” indicates *m* and *n* of the average feature values in the freshly preserved and cryo-preserved MSC groups, respectively. *P_uni_* and *P_adj_* indicate *p*-values that were evaluated by using univariate and multivariate (adjusting age) linear regression analyses; paracrine molecules (VEGF, MCP1, and IL6) and the preservation method were set as the dependent and independent variables, respectively. The median and interquartile range are described in Appendix A. Abbreviations: VEGF, vascular endothelial growth factor; MCP1, monocyte chemoattractant protein 1; IL6, interleukin 6.

**Figure 8 cells-12-02355-f008:**
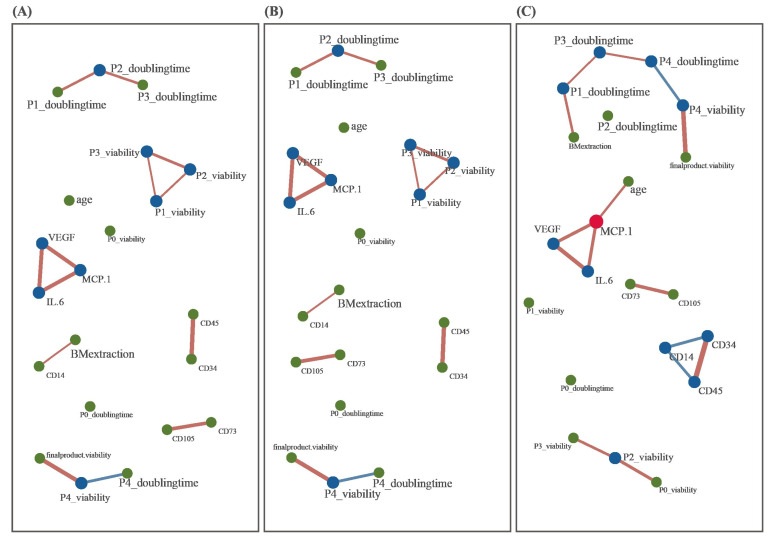
Network analysis using 21 variables of autologous BM-MSCs. Correlation matrices were produced by using all cases (**A**), freshly preserved BM-MSCs (**B**), and cryo-preserved BM-MSCs (**C**). Each correlation matrix includes the values of the PCC between all possible pairs of features (n = 21), yielding a 21-by-21 matrix. The network was constructed by using feature interactions with a PCC of 2 or higher and the igraph package [34]. Green-, blue-, and red-colored points indicate nodes with a number of edges of zero to one, two, and three, respectively. Red and blue lines denote positive and negative PCCs, respectively. The line width is proportional to the absolute PCC value. Abbreviations: BM, bone marrow; VEGF, vascular endothelial growth factor; MCP1, monocyte chemoattractant protein 1; IL6, interleukin 6.

**Table 1 cells-12-02355-t001:** Therapeutic application of autologous BM-derived MSCs in human diseases.

Study	Disease	Year	Preservation	Injection Route	Therapy Times	Design
Bang et al. [15]	Stroke	2005	Fresh	Intravenous	2 times	RCT
Lee et al. [16]	Multiple system atrophy	2008	Fresh	Intra-arterial (ICA) + Intravenous	2 times	RCT
Lee et al. [17]	Stroke	2010	Fresh	Intravenous	2 times	Open-label trial
Park et al. [18]	Spine	2011	Fresh + cryo	Intramedullary (spine)	1 time	Case study
Lee et al. [19]	Patients receiving living donor KT	2013	Not traceable	Intra-osseous	1 time	Open-label trial
Lee et al. [20]	MI	2014	Not traceable	Intra-arterial (coronary artery)	1 time	RCT
Jang et al. [21]	LC, alcoholic	2014	Fresh + cryo	Intra-arterial (hepatic artery)	2 times	Open-label trial
Oh et al. [22]	Spine	2015	Not traceable	Intramedullary (spine)	1 time	Case study
Suk et al. [23]	LC, alcoholic	2016	Fresh + cryo	Intra-arterial (hepatic artery)	1 time + 2 times	RCT
Kim et al. [24]	MI	2018	Not traceable	Intra-arterial (coronary artery)	1 time	RCT
Kim et al. [25]	LC, HBV	2021	Fresh	Intra-arterial (hepatic artery)	2 times	Case study
You et al. [26]	Erectile dysfunction	2021	Not traceable	Corpus cavernosum	1 time	Open-label trial

Abbreviations: BM-MSC, bone-marrow-derived mesenchymal stem cell; RCT, randomized clinical trial; ICA, internal carotid artery; KT, kidney transplantation; MI, myocardial infarction; LC, liver cirrhosis; HBV, hepatitis B virus.

## Data Availability

Not applicable.

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
