# Peer review of "Comparative Analysis of Biological Signatures between Freshly Preserved and Cryo-Preserved Bone Marrow Mesenchymal Stem Cells"

_cells, 2023, doi:10.3390/cells12192355_

Round 1

Reviewer 1 Report (Previous Reviewer 3)

All of my issues were sufficiently adressed.

Author Response

Thank you for the positive assessment despite my research still being in its nascent stages.

Reviewer 2 Report (Previous Reviewer 2)

  The authors submitted a revised manuscript and made some corresponding changes highlighted based on the reviewers' comments. Any change, even a letter or a punctuation mark, should be highlighted for a revised manuscript. It is a waste of time and effort for me to compare the differences between the previous and revised versions of the manuscript, as the authors only highlighted some changes in response to reviewers' comments and left the other changes unmarked. Although the authors mentioned “We have received overall English proofreading once again, and l have made corrections to all the points you pointed out.” in the response, the revised manuscript still contains some errors.

  Although the authors mentioned “We have received overall English proofreading once again, and l have made corrections to all the points you pointed out.” in the response, the revised manuscript still contains some errors.

Author Response

We apologize for diluting the efforts you put into evaluating our research by not highlighting all the changed parts adequately. We carried out the English editing again with the help of a professional editing team (MDPI English Editing). We have tried to alleviate your burden by marking all changed parts as possible, comparing it to the text at the time of the initial submission.

Reviewer 3 Report (Previous Reviewer 1)

The submitted work is well-presented, and the addition of new paragraphs improves the overall scientific soundness. Despite the fact that it was noted in the discussion that this work is statistical, this information should be explicitly stated in the introduction as well as in the appropriate paragraph of the materials and methods.

Minor English editing may be introduced.

Author Response

Thank you for the positive assessment and minor revision. We response your comments with a PDF file.

Reviewer 4 Report (New Reviewer)

Paper entitled "Comparative analysis of biological signatures between freshly preserved and cryo-preserved bone marrow mesenchymal stem cells" by Lee et al. outlines biological characteristics of fresh and cryopreserved MSCs. Study is well performed and written, obtained results are significant for MSCs therapy and other applications.  Overall, I recommend to accept proposed paper in form as it is.

Author Response

Thank you for reviewing our research in detail.

Thanks once again for the favorable assessment.

Round 2

Reviewer 2 Report (Previous Reviewer 2)

The authors submitted a revised manuscript and made corresponding changes highlighted based on the reviewers' comments. 

Very few English errors need to be corrected.

This manuscript is a resubmission of an earlier submission. The following is a list of the peer review reports and author responses from that submission.

Round 1

Reviewer 1 Report

Topic presented in the paper of Lee et al. is very interesting, but major changes should be introduced in order to enhance significance of the paper.

·         The procedures are described in an unorganized way. The paper should be written in a way that is clear also to novices who will be doing their own stem cell studies. What samples were taken from the Pharmicell database? Are they the 671 samples that went through cell culture? You conducted analysis on 2300 cases contained in the Pharmicell database or on 671 cultivated samples?

·         In what media were the cells cultured? In what medium were the samples kept throughout freezing or room temperature storage? As demonstrated in the research of ÅšcieżyÅ„ska A, SoszyÅ„ska M, Szpak P, KrzeÅ›niak N, Malejczyk J, KalaszczyÅ„ska I. Influence of Hypothermic Storage Fluids on Mesenchymal Stem Cell Stability: A Comprehensive Review and Personal Experience. Cells. 2021, these factors have a significant impact on the viability and stemmness of MSC. How long were the samples kept in the LN2 tank? General facts on the cell culture procedure should be clearly communicated.

·         You counted cell viability with a hemocytometer under a light microscope, which might lead to mistakes. How many repetitions did you do for counting?

·         What kind of variables were chosen, and who were the experts listed in your publication that selected them?

·         There are much too many extra figures. This makes it difficult to switch from the main text to the additional figures.

·         Which website was used to create Figure1?

Minor editing of English language required

Reviewer 2 Report

The authors submitted a manuscript investigating the comparison of biological signatures between freshly preserved and cryo-preserved MSCs.

Technologies to cryogenically preserve living tissue, cell lines and primary cells have matured greatly for both clinicians and researchers since their first demonstration in the 1950s and are widely used in storage and transport applications. Cryopreservation represents an efficient method for the preservation and pooling of MSCs, to obtain the cell counts required for clinical applications, such as cell-based therapies and regenerative medicine. Upon cryopreservation, it is important to preserve MSCs functional properties including immunomodulatory properties and multilineage differentiation ability.

Based on the current status of the widespread use of cell cryopreservation, it makes little sense for the authors to compare freshly preserved MSCs with cryo-preserved MSCs, not to mention that it can't be called freshly preserved MSCs, and the authors are not comparing them by specific experiments but based on data from databases. For another thing, the effects between those used for different disease treatments can't be directly compared in this way.

The English needs to be improved to a certain extent. There are some errors in grammar and format in the whole manuscript: inconsistencies; tense; single and plural expressions; the use of prepositions and definite/indefinite articles; punctuation. For example:

IL-6, IL6;

In Line 70, “such stroke” should be changed into “such as stroke”;

In Line 173, “2.3” should be changed into “2.4”;

The English needs to be improved to a certain extent. There are some errors in grammar and format in the whole manuscript: inconsistencies; tense; single and plural expressions; the use of prepositions and definite/indefinite articles; punctuation. For example:

IL-6, IL6;

In Line 70, “such stroke” should be changed into “such as stroke”;

In Line 173, “2.3” should be changed into “2.4”;

Reviewer 3 Report

The manuscript entitled „Comparative analysis of biological signatures between freshly preserved and cryo-preserved bone marrow mesenchymal stem cells” by Taesic Lee et al is of high interest for the readers of Cells.

The authors compared  the influence of freezing on the quality (viability, population doubling time, cytokine secretion) of MSC which were either frozen and thawed or being directly cultivated. A large data base was available for evaluation and sophisticated statistical approaches were undertaken to search for differences between the two groups. Almost no differences were observeable and the authors conclude that ”the biosignatures of cryo-preserved BM-MSCs were highly conserved” compared to unfrozen MSCs.

Since stem cell therapies using autologous or (upcoming) allogenous MSC gain more and more attention and clinical relevance for treatment of increasing number of different pathologies, the question if cryo-preserved or fresh cells will be better is of great interest in this area. The large number of parameters assessed for quality control of MSCs as well as the large cohorts allow a robust statistical comparison between both groups.

A well written manuscript is provided, the methods were presented comprehensively and the results were presented and discussed adequately. However, some questions remain as pointed out in the following.

1.) Several parameters were assessed but I miss data on trilineage differentiation which is also important criteria, if MSC were functionally tested. However, these data would be a nice add on but not necessary for acceptance in my opinion. It is published that the differentiation potential of MSCs drops significantly with increasing passage number. The data presented in the manuscript at least suggest that this might happen in the 4th passage when the viability starts to decrease and population doubling time increases. Both were signs of cellular senescence.

2. Over what period of time were the MSCs stored in nitrogen? Please provide a mean time.

3. Please provide some informations about the freezing medium that was used.

4. Please provide a correlation analysis between volume of bone marrow and functional parameters (especially with population doubling time at P4). It might be possible, that the MSC need more population doublings before they can used the first time because initial total number is low if the volume of bone marrow is low. If there is a positive correlation please add this aspect to the manuscript. If not, keep it as it is.

5. Introduction:  Page 2 l55. You state here that Bone marrow as potential MSC source is difficult to obtain (which is somehow true). But in the following you mention organs like pancreas or liver as potential sources. I am sure that it is much more complicated to obtain MSC from these internal organs compared to BM-aspiration from iliac crest. Please revise accordingly.

6. Materials and Methods: Please indicate, if the MSC were used autologously or allogenic, preferentially in section 2.1.

7. Materials and Methods: Please describe in more details how cytokines were collected (timeframe in culture, was there any stimulation necessary?).